# Silicon Nanowire Field-Effect Transistor as Biosensing Platforms for Post-Translational Modification

**DOI:** 10.3390/bios10120213

**Published:** 2020-12-21

**Authors:** Ping-Chia Su, Bo-Han Chen, Yi-Chan Lee, Yuh-Shyong Yang

**Affiliations:** 1Department of Biological Science and Technology, National Chiao Tung University, Hsinchu 300, Taiwan; Kisky0908@hotmail.com (P.-C.S.); J730703@yahoo.com.tw (B.-H.C.); st30233lee@gmail.com (Y.-C.L.); 2Center for Intelligent Drug Systems and Smart Bio-Devices (IDS2B), National Chiao Tung University, Hsinchu 300, Taiwan

**Keywords:** polycrystalline silicon nanowire field-effect transistor (pSNWFET), post-translational modifications (PTMs), protein tyrosine sulfation (PTS), protein–protein interaction

## Abstract

Protein tyrosine sulfation (PTS), a vital post-translational modification, facilitates protein–protein interactions and regulates many physiological and pathological responses. Monitoring PTS has been difficult owing to the instability of sulfated proteins and the lack of a suitable method for detecting the protein sulfate ester. In this study, we combined an in situ PTS system with a high-sensitivity polysilicon nanowire field-effect transistor (pSNWFET)-based sensor to directly monitor PTS formation. A peptide containing the tyrosine sulfation site of P-selectin glycoprotein ligand (PSGL)-1 was immobilized onto the surface of the pSNWFET by using 3-aminopropyltriethoxysilane and glutaraldehyde as linker molecules. A coupled enzyme sulfation system consisting of tyrosylprotein sulfotransferase and phenol sulfotransferase was used to catalyze PTS of the immobilized PSGL-1 peptide. Enzyme-catalyzed sulfation of the immobilized peptide was readily observed through the shift of the drain current–gate voltage curves of the pSNWFET before and after PTS. We expect that this approach can be developed as a next generation biochip for biomedical research and industries.

## 1. Introduction

Protein tyrosine sulfation (PTS), a common post-translational modification (PTM), regulates numerous physiological responses and pathological diseases [1,2,3]. The occurrence of PTS was estimated to be up to 1% of total tyrosine and approximately 30% of proteins [3]. In our previous study, we found that among 4256 *Escherichia coli* proteins, 875 sulfated proteins were identified using a PTS system and proteome microarray [4]. PTS participates in some protein–protein interactions including hemostasis, leukocyte rolling on endothelial cells, ligand binding to receptors, visual functions, inflammatory responses, and viral entry into host cells [1,2,5,6,7,8,9]. PTS was reported to facilitate infection with the human immunodeficiency virus (HIV) and enterovirus (EV) 71 [2]. PTS of C–C chemokine receptor type 5 (CCR5) is involved in the entry of HIV-1 into CD4+ cells. The sulfation site of CCR-5 is N-terminal tyrosine. The entry of HIV-1 into CCR5- and CD4-expressing cells is facilitated by sulfated tyrosine, which contributes to the binding of macrophage inflammatory protein (MIP)-1α and MIP-1β to HIV-1 gp120/CD4 complexes. EV71 is strongly associated with sulfated P-selectin glycoprotein ligand (PSGL)-1 [10]. The infection of leukocytes with EV71 is highly correlated with N-terminal tyrosine sulfation of PSGL-1 on leukocytes. The capsid protein VP1 of EV71 binds to sulfated tyrosine residues of PSGL-1, thus facilitating the entry of EV71 into cells. Strains of EV71 can be distinguished as EV71–PSGL-1-binding (PB) and EV71–non-PB strains. EV71–PB strain infections can lead to fatal encephalitis and hand, foot, and mouth disease [11].

Monitoring PTS is difficult owing to the instability of sulfated proteins and the lack of direct real-time detection methods [12,13]. The most frequently used method for PTS detection is the incorporation of the radioactive isotope, Sulfur-35 (35S), which can confirm PTS by radiolabeling the protein substrate [14]. It is a direct and highly sensitive method for detecting PTS. However, the instability of the sulfate organic ester may cause variations during sample preparation [15]. High-resolution Fourier transform mass spectrometry (MS) is used for the direct identification of the sulfation site of secreted peptides [16]. Although MS is a powerful tool for studying PTMs, its application in the analysis of sulfotyrosines remains challenging, as false negative results may be attributed to sulfate groups potentially decomposing in negative and positive ions in tandem MS experiments [17,18]. Monoclonal antibodies that can specifically recognize sulfotyrosines are typically used in the detection of PTS owing to their high specificity and ease of incorporation into traditional immunoassays [19]. However, this is a less sensitive and label-based method that requires indirect signal amplification. Mass and two-dimensional gel electrophoresis, which are powerful techniques [20], were developed for conducting PTM functional proteomics studies, particularly phosphoproteomics [21,22]. However, these powerful techniques have not yet been reported in sulfoproteomic research studies. A sensitive method that enables direct identification of PTS in real time and that can be developed for high-throughput applications is essential. A real-time method for the sulfation of a peptide has been developed using a fluorescent labelled peptide and microfluidic system to separate the sulfated product and the substrate. The synthetic peptide with fluorescent labeling is needed for this assay [23]. In this study, we used a peptide that contained the PSGL-1 sulfation site to study PTS by using a polysilicon nanowire field-effect transistor (pSNWFET) to directly monitor the immobilized sulfated peptide on the nanowire (NW) of the transistor.

In the last decade, SNWFETs function as biosensing transducers that can be used in the ultrasensitive monitoring of charged biomolecules, and they can be transformed into electrical signals in real time. On the basis of the sensing principle of the NWFET, many new applications and advantages have begun to emerge for the nanosized NWFET. Because of its large surface-to-volume ratio, the sensitivity of the device can meet the requirements of biological detection. Biomolecules such as nucleic acids, antibodies, and proteins can be chemically modified to the NW in a simple manner as per requirements [24]. A simple and efficient method for fabricating pSNWFETs for biosensing applications was further demonstrated using the polycrystalline silicon sidewall spacers etching technique, which is compatible with complementary metal oxide semiconductor (CMOS) processes [25]. A field-effect transistor (FET) can be mass produced in array types and with integrated circuits, which may become a platform for a sensitive, label-free, real-time, and high-throughput biosensing system. Such a system would be the most suitable for various omics applications, such as PTMs of proteins. Artificial intelligence (AI) has become very popular and is changing the way in which we interact with technology. In the field of health care, AI can collect and provide extensive data to determine treatment options for individuals with genetic abnormalities. A semiconductor-based biochip can be easily associated with the digital world such as Cloud and AI, making it a powerful tool for industrial applications.

Due to the maturity of the semiconductor process, silicon-based nanowire FET device for biosensing application has been widely studied for the past two decades. At present, other FET-based biosensing devices are also thriving in academic research, for example, carbon nanotube and graphene FET are shown to exhibit excellent properties for biosensing applications [26,27]. Such emerging FET devices may provide many novel opportunities for the future development of next generation biochips. Biosensors using the surface of graphene has many advantages such as high carrier mobility, low intrinsic electrical noise, and inert chemical properties. However, there are still challenges to overcome for the development of graphene, for instance, it does not have a band gap, which means it cannot be switched off [26]. Scientists are still working on improving the graphene-based semiconductor. Once graphene FET overcomes those shortcomings, it will provide many opportunities for biosensing application in the future.

We previously demonstrated that the proteome microarray could rapidly and efficiently find potential proteins that were subjected to tyrosine sulfation [4]. Further, through the use of atomic force microscopy (AFM), it was discovered that tyrosylprotein sulfotransferase (TPST)-induced protein–protein interactions were mainly caused by the sulfation of Y-51 within PSGL-1 [28,29]. These results suggested that PSGL-1 is a PTS substrate, and self-assembled monolayers (SAMs) can be applied to immobilize proteins onto the silicon surface; PTS reactions can be conducted on the silicon surface. Herein, we demonstrated the monitoring of in situ PTS and its subsequent protein–protein interactions by using a pSNWFET. Further, we confirmed that PTS can be detected in real time by using the pSNWFET through simple steps. The results showed that the pSNWFET exhibits great potential as a tool for PTS detection and screening in high-throughput applications, providing a simple and direct method for monitoring PTS in real time for sulfoproteomic studies in the future.

## 2. Materials and Methods

### 2.1. Materials

Ethanol (99.8%), 3-aminopropyltriethoxysilane (APTES), glutaraldehyde, sodium cyanoborohydride, bis-tris propane, Tween-20, adenosine 3′-phosphoadenosine 5′-phosphosulfate (PAPS), adenosine 3′,5′-diphosphate (PAP), 3,3′5,5′-tetramethylbenzidine (TMB), 2-mercaptoethanol, 2-(N-morpholino)ethanesulfonic acid (MES) hydrate, 4-methylumbelliferone (MU), MU sulfate (MUS), and anti-mouse IgG (whole molecule)–Gold antibody were obtained from Sigma (St. Louis, MO, USA). Monobasic and dibasic sodium phosphate were obtained from JT Baker (Phillipsburg, NJ, USA). The PSGL-1 peptide ATEYEYLDYDFL and the sulfated PSGL-1 peptide ATEYEYLDYsDFL were synthesized by Genemed Synthesis Inc. (San Antonio, TX, USA) and Kelowna International Scientific Inc. (Taiwan), respectively. Antisulfotyrosine antibody and horseradish peroxidase (HRP)-conjugated mouse immunoglobulin G (IgG) antibody were obtained from Millipore and Abcam, respectively. Recombinant phenol sulfotransferase (PST) (K65ER68G of rat PST) and *Drosophila melanogaster* tyrosylprotein sulfotransferase (DmTPST) were expressed by *E. coli* by using vectors pET-3c, pGEX-4T1, and pET-43a, according to previously described procedures [13,28].

### 2.2. Fabrication of pSNWFET

The pSNWFET was fabricated at National Nano Device Laboratories (Hsinchu, Taiwan) by using a previously reported sidewall spacer technique [25]. In the beginning, a 100 nm-thick thermal oxide layer and a 50 nm-thick silicon nitride layer were deposited on a p-type 6-inch Si wafer by chemical vapor deposition (CVD), followed by a 100 nm-thick TEOS oxide deposition and photo lithography to define the dummy structure. Next, a 100 nm-thick amorphous-Si layer was deposited by LPCVD and then an annealing process was performed to transfer α-Si into polycrystalline structure. Subsequently, source/drain (S/D) implant was accomplished by P31^+^. A reactive plasma etching step was subsequently performed to remove the poly-Si layer. Finally, the sidewall poly-Si NW channels were formed due to the sidewall spacer technique. The n-type poly-Si NWs serving as one-dimensional conducting channels were fabricated using completely complementary metal–oxide–semiconductor-compatible processes. This method is compatible with CMOS technologies, making mass production a reality. The contact material is doped polysilicon. The width of the poly-Si NWs is 80 nm and the diameter of the NWs is 50 nm. The structure of poly-Si FET was illustrated in Scheme 1.

### 2.3. Electrical Measurements of pSNWFET

All measurements were conducted at room temperature (RT). The sensor was characterized according to its drain current (I_D_)–gate voltage (V_G_) response measured using a commercial semiconductor analyzer (Keithley 2636). I_D_ was measured at a constant drain voltage of 0.5 V while increasing V_G_ from −1 to 2 V. Each I_D_-V_G_ curve was measured thrice. A gate voltage shift (ΔV_G_) was obtained by calculating the differences in the gate voltage at a drain current of 10^−8^ A. The electrical response for each step was measured thrice, and the average value was reported. Alternating current (AC) conductance was measured to determine the real-time variations in electrical signals. A gold microwire served as the liquid gate electrode, and an AC signal generator (SR830 DSP Lock-In Amplifier) was used. A sine wave (76 Hz; amplitude 30 mV) was applied to the drain. The measured conductance of the device increased as the voltage of the liquid-gate electrode increased from 0 to 1.3 V in the buffer solution.

### 2.4. Electrical Characteristics of pSNWFET

The electrical characteristics of pSNWFET were determined under aqueous conditions. The drain current-gate voltage curve (I_D_-V_G_) was measured, as shown in Appendix A. The performance of the device shows high on/off ratio (around six orders) and low gate leakage current. The I_D_-V_G_ curve remained stable in eight consecutive measurements, indicating the stability of the pSNWFET, as shown in Appendix A.

### 2.5. Immobilization of PSGL-1 Peptide on the NW Surface

The immobilization procedure is illustrated in Scheme 2. The as-fabricated device was immersed in acetone and ethanol for 5 min to remove surface contamination, followed by oxygen plasma treatment for 5 min. The device was subsequently immersed in 2% APTES–ethanol solution for 30 min and heated at 120 °C for 10 min to produce amine groups on the Si NWs. The NW device was subsequently incubated with 12.5% glutaraldehyde and 10 mM pH 7 phosphate buffer (PB) for 1 h. The device was treated with PSGL-1 peptide (ATEYEYLDYDFL) (10 μM in a PB solution) for 12 h in order to attach the peptide to the NW surface. Finally, the device was immersed in 10 mM Tris-HCl (pH 7.0) with 4 mM sodium cyanoborohydride for 10 min to block unreacted functional groups and prevent subsequent nonspecific interactions. The immobilization of synthesized sulfated PSGL-1 peptide (ATEYEYLDYsDFL) was conducted in the same manner as the aforementioned method.

### 2.6. Enzyme-Catalyzed Tyrosine Sulfation of the Immobilized PSGL-1 Peptide Substrate

As shown in Scheme 2, a PST–TPST coupled enzyme system was developed to determine TPST activity and produce PTS in our previous study [13,28]. In this study, the system was used to catalyze the sulfation of the immobilized PSGL-1 peptide on an enzyme-linked immunosorbent assay (ELISA) plate and the poly-Si NW. The reaction mixture comprised 50 mM MES buffer (pH 6.5), 5 mM 2-mercaptoethanol, 30 μM PAPS, 2 mM MUS, 17 mU (28 μg) K65ER68G PST, and 25 μg of NusA–DmTPST in a volume of 200 μL. This reaction mixture was incubated at 37 °C for 30 min, and 10 mM bis-tris propane buffer was added at the end of the reaction. TPST catalyzed the PTS of PSGL-1 peptide, which was both immobilized on the ELISA plate and the NW surface for subsequent sensing procedures.

### 2.7. Preparation for Scanning Electron Microscopy

The immobilized device was treated with gold nanoparticle-conjugated antibodies to determine the successful immobilization of the biomolecule on the NW surface [29]. PSGL-1 peptide was immobilized onto the NW surface and subsequently treated with PST–TPST coupled enzyme to produce sulfated PSGL-1. Anti-sulfotyrosine antibody was subsequently treated, and the gold nanoparticle-conjugated antibody was used as the reporter. For the control group, the aforementioned experiment was performed in the absence of TPST. Scanning electron micrographs were obtained from Integrated Service Technology (Hsinchu, Taiwan). Anti-mouse immunoglobulin (IgG) conjugated with 10-nm gold nanoparticles was used as the label for detection through scanning electron microscopy (SEM). The device was immersed in 250 ng/mL anti-sulfotyrosine antibody in 10 mM PB for 2 h and washed thrice with 0.05% Tween-20 to eliminate nonspecific binding. Finally, the surface of the device was submerged in mouse IgG (1:1000) in 10 mM bis-tris propane for 3 h and washed thrice with 0.05% Tween-20.

### 2.8. X-ray Photoelectron Spectroscopy

X-ray photoelectron spectroscopy (XPS), electron spectroscopy for chemical analysis, was used for determining the functionalization of the NW surface [29]. XPS measurement was performed using a Thermo Fisher Scientific Theta Probe with a monochromatic X-ray source (Al Kα, 1486.6 eV) and a concentric hemispherical analyzer. Binding energy of 284.6 eV for the C 1s peak was calibrated as the reference. XPS analysis was performed by National Nano Device Laboratories (Hsinchu, Taiwan).

### 2.9. ELISA-Based Detection of PTS

PSGL-1 peptide (ATEYEYLDYDFL) was coated on the amine-binding ELISA plate at 37 °C for 2 h in phosphate-buffered saline (PBS). Subsequently, to block the remnant amine binding sites, 3% milk was added to PBS at RT for 1 h. PTS on the immobilized PSGL-1 was catalyzed by the PST–TPST coupled enzyme system, followed by washing with 0.05% Tween-20/PBS thrice. The reaction mixture was incubated with primary (a mouse anti-sulfotyrosine antibody diluted with PBS, 1:1000) and secondary antibodies (an anti-mouse HRP-conjugated antibody, 1:5000) at RT for 1 h and 30 min, respectively. After each interval of the experimental procedure, the ELISA plate was washed with PBS/0.05% Tween-20 for three times. Finally, 100 μL of tetramethylbenzidine substrate for HRP was added and incubated at RT for 30 min. The HRP-catalyzed reaction was terminated using 100 μL of 2 M H_2_SO_4_, and the resulting chemiluminescence was quantified at OD450 nm by using an ELISA reader (VICTORTM X3; Perkin Elmer, Waltham, MA, USA).

## 3. Results and Discussion

### 3.1. Surface Modification and Verification of pSNWFET

Appropriate immobilization of PSGL-1 peptide or the sulfated form of the peptide on the sensing surface of the pSNWFET is critical for successful PTM monitoring by the device. XPS and SEM were used to confirm the surface modification on the pSNWFET in this study.

SEM was used to observe gold nanoparticle-conjugated anti-mouse IgG antibodies. The gold nanoparticle-conjugated anti-mouse IgG antibodies were designed to bind to the anti-sulfotyrosine antibody. The surface morphology of the functionalized pSNWFET, as observed on SEM, is shown in Figure 1. PSGL-1 peptides were first immobilized on the pSNWFET surface according to the aforementioned steps in the Materials and Methods section. By applying the PST–TPST coupled enzyme system, the PTS reaction of the tyrosine residue of PSGL-1 was performed, and sulfated PSGL-1 was then generated through TPST, as described in the Materials and Methods section. The anti-sulfotyrosine antibody is specific for the recognition of sulfated peptides and was used to confirm the presence of sulfated PSGL-1. The gold nanoparticle-conjugated anti-mouse IgG antibody, which in turn recognizes the anti-sulfotyrosine antibody, was subsequently used to cover the pSNWFET surface. As shown in Figure 1A, gold nanoparticles (10–12 nm) were observed in SEM images. However, as shown in Figure 1B, gold nanoparticles were not observed when the critical enzyme TPST was absent in the PST–TPST coupled enzyme system. The results verified that the surface modification and the in situ PTS on the pSNWFET surface were successful.

Further evidence for successful surface modification on poly-Si surface was obtained through XPS (Figure 2), which is an elemental analysis technique widely used to characterize the surface chemistry of materials used in bioengineering. Figure 2A–C represent the XPS spectra of oxygen, carbon, and nitrogen, respectively, during each immobilization step. The XPS peak of carbon and nitrogen is generally used as an indicator of organic molecule or protein deposition [30]. Previous studies have reported that the C 1s, O 1s, and N 1s spectra change under different modifications or conditions [31,32]. As oxygen is also on the poly-Si surface, significant variation of oxygen XPS spectra could be observed only after the PSGL-1 peptide was immobilized (Figure 2A). The characteristics of the C 1s spectra before and after the functionalization clearly indicated polypeptide immobilization on the poly-Si surface (Figure 2B). The C 1s peak of nude was a background value. Further, the N 1s spectrum showed an apparent peak at a binding energy of 402 eV, which corresponded to the numerous nitrogen atoms within the polypeptide. Appendix A shows the content of the surface elements of the modified surface, corresponding to the previous results. The results indicate that the PSGL-1 peptide was successfully immobilized on the pSNWFET surface through APTES and glutaraldehyde as linker molecules for biosensing application.

The aforementioned findings showed that the biochemical reactions observed corresponded to those of a coupled enzyme system for the sulfation of PSGL-1 peptide, which were performed on the semiconductor surface of the pSNWFET. The silicon oxide surface is as common as glass as a substrate for biochips [33]. Similar procedures can be used for the immobilization of probes on silicon NWs that contain either native or artificially grown silicon oxide surface. Functional groups, such as amine, can be processed directly in the foundry right after wafer tape-out. Such fabrication of a semiconductor-based biosensor can be even more efficient than the current method for the immobilization of probes on traditional biochips. Various functional groups can be modified on the pSNWFET surface and can be easily examined through the change in the electric properties of the pSNWFET. In this study, we further demonstrated that enzyme-catalyzed PTM can also be performed on the pSNWFET and verified such PTM.

### 3.2. Identification of In Situ PTS with ELISA

The aforementioned experiments were designed for the verification of immobilization on the pSNWFET. For the functional verification of PTS, ELISA was used as a parallel experiment to observe and confirm in situ PTS in solution and on the pSNWFET, as shown in Figure 3. ELISA is a convenient method that can be used to determine the sulfated protein [28]. The PTS reaction monitored through ELISA was similar to that described in Scheme 2 for PTS on the pSNWFET, with minor modifications. Scheme 1 shows the structure and the function of the electronic device and biosensing application. Figure 3A illustrates that the sulfated peptide can only be obtained through a complete enzyme mixture with the PST–TPST system. In the absence of the critical enzyme TPST or the substrate peptide (PSGL-1), negative results were obtained. The results indicate that the sulfated peptide could only be obtained through the PTS-TPST coupled enzyme system and could be recognized by anti-sulfotyrosine antibody (Figure 3A). The results revealed that in situ PTS could be achieved through coupled enzyme treatment, and this was used as supporting data for the following experiment in which in situ PTS was catalyzed on the pSNWFET surface.

In addition, we utilized ELISA to demonstrate protein–protein interactions at different buffer concentrations (Figure 3B). The concentration of the sensing buffer plays an important role and affects the performance of the FET [34]. Low-salt buffer is generally preferred for pSNWFET base biosensing to maintain a longer Debye length [34]. However, ionic strength also had a strong influence on the binding of antigen by the corresponding antibody and enzyme-catalyzed reactions, as shown in Figure 3B. It is suggested that a 10 mM concentration of bis-tris propane buffer, which is appropriate for the pSNWFET base biosensor, is also suitable for the PTS coupled enzyme system and related antibody/antigen interactions.

The principle of ELISA is exactly similar to that of our semiconductor-based protein sensing, except that the sensitivity can vary around 5 to 6 orders of magnitude [35], because the former usually relies on fluorescence labeling, and the latter can be directly monitored though the charge interaction between the target molecules and the NWs. In our study, ELISA was used as a comparative tool to first establish the optimal biosensing condition for our method (Figure 3). Any potential biosensing conditions for the semiconductor-based protein sensing were examined thoroughly using ELISA to confirm that expected biochemical reactions occurred, and the conditions were then selected for further experiments.

### 3.3. Electrical Responses of the Functionalized pSNWFET after Each Step of the Modification Process

Surface alterations on the pSNWFET can be monitored through its electrical responses, which can be used to further confirm the surface functionalization at each modification step. Figure 4 illustrates the electrical response at each modification step. Following functionalization of amines on the n-type pSNWFET device through APTES (M2/APTES in Figure 4), a left shift from the baseline (M1/Bare in Figure 4) was observed. The increase in I_D_ was induced by the positive charges on the NW surface after APTES modification. Subsequently, the surface was functionalized by an aldehyde group following glutaraldehyde treatment. I_D_ decreased (M3/glutaraldehyde in Figure 4) because of the formation of imine and elimination of positive charges. Finally, the device was modified with the PSGL-1 peptide on the surface, and electrical measurement was performed, resulting in a considerable decrease in I_D_ (M4/PSGL-1 in Figure 4). The shift could be attributed to the presence of negative charges within the PSGL-1 polypeptide, which contains four acidic amino acids. As shown in Figure 4, at each modification step, the I_D_-V_G_ curves shifted according to the SiNW surface charge. This result indicates that a sensitive pSNWFET device for assessing the surface properties of proteins with different modifications can be developed for biosensing application.

With regard to data collection, transmission, and processing, distinct differences exist between the traditional biochip and the semiconductor-based pSNWFET biochip. pSNWFET chip fabrication in a modern semiconductor foundry is CMOS-compatible, and the electronic signals obtained can be directly processed through well-established electronics and communication industries. Further, the mass production and high quality of the pSNWFET chips can be facilitated by modern electronic techniques. Reproducibility of the academic chip fabrication process has been the main hindrance in its biomedical application, because the parameters of the fabrication process are not fixed, and reproducibility is usually uncertain. Such difficulty can be clearly eliminated with commercial processes. Our data (Appendix A) demonstrated that reproducible and unambiguous biosensing data can be obtained using chips fabricated in an academic foundry (NDL). Recently, such processes have been successfully transferred to commercial foundries, resulting in considerably the improved yield and stability of pSNWFET chips (data not shown).

### 3.4. Electrical Responses of PTS and Antibody Recognition on pSNWFET

The pSNWFET was developed for biosensing applications including nucleic acids, cancer biomarker proteins, and other small molecules; it has been proven that the pSNWFET has the potential to serve as a platform for the detection of biomolecules [24]. In this report, we would like to demonstrate further, using PTS in situ protein modification as an example, that biochemical reactions can be directly monitored and confirmed on the pSNWFET. In the previous section, we demonstrated that on-chip PTS could be catalyzed by the PST–TPST coupled enzyme system, which was verified by SEM and ELISA as parallel experiments. Moreover, we demonstrated that enzyme-catalyzed protein sulfation on the NW surface was feasible and could be detected through real-time electrical measurement. The electrical response and specificity of the functionalized pSNWFET for PSGL-1 sulfation sensing are shown in Figure 5. As shown in Figure 5A, the sulfated PSGL-1 peptide (sulfated at the third tyrosine residue, ATEYEYLDYsDFL) was immobilized on the NW surface, and the electronic responses after the addition of antibodies were recorded. The I_D_-V_G_ curve remained unchanged in the presence of noninteracting biomolecules (anti-glutathione-S-transferase antibody, M2, which serves as negative-controlled antibody to confirm that nonspecific binding does not occur with antibody other than the anti-sulfotyrosine antibody, in Figure 5A), indicating that the electrical condition of the pSNWFET was stable in the presence of nonspecific molecules. This is an important characteristic, as many nonspecific molecules that may carry charges exist in biological samples. The anti-sulfotyrosine antibody treatment resulted in reduced I_D_ (M3 in Figure 5A), owing to the interaction between sulfated PSGL-1 and the corresponding high-affinity antibody, as proven in the previous section (Figure 3). This result showed that the sulfated peptide could be specifically detected on the pSNWFET surface by the anti-sulfo antibody and was consistent with that of the ELISA experiment (Figure 3). To normalize current alteration in electrical detection, a shift of gate voltage (ΔV_G_, shown in the inset figure) at I_D_ = 10^−8^ A was calculated and is illustrated in Figure 5. The average values of ΔV_G_ between base line/nonspecific antibody (M2-M1 in Figure 5A) and base line/anti-sulfo antibody (M3-M1 in Figure 5A) were −5 mV and 57 mV, respectively, as shown in Figure 5B. Figure 5A shows one of the four typical experiments, and the data of the other three experiments are shown in Appendix A.

### 3.5. Monitoring Enzyme-Catalyzed PTS on pSNWFET

As shown in Figure 6A, after the immobilization of the nonsulfated PSGL-1 peptide onto the pSNWFET surface (M1 in Figure 6A), the PST–TPST coupled enzyme treatment was conducted to generate PTS (M2 in Figure 6A). A small but crucial increase in the gate voltage shift was observed owing to the presence of the negatively charged sulfate group (ΔV_G_ = 20 mV at 10^−8^ A drain current from M1 to M2, inset of Figure 6A). Incubation of anti-GST antibody, which does not recognize the sulfated peptide, was used as negative control, and the electrical response remained constant (M3 in Figure 6A). Addition of the anti-sulfotyrosine antibody induced a significant change in the electrical response (ΔV_G_ = 61 mV at 10^−8^ A drain current from M3 to M4). For the control experiments, as shown in Figure 7A, the natural nonsulfated PSGL-1 peptide was immobilized and measured as the baseline (M1), as shown in Figure 6A. However, TPST, the enzyme responsible for the catalysis of PTS, was removed from the PST–TPST coupled enzyme treatment; as a result, the peptide was lacking the sulfate group (M2 in Figure 7A). As expected, the I_D_-V_G_ curves remained unchanged even after incubation with anti-GST (M3 in Figure 7A) or anti-sulfotyrosine (M4 in Figure 7A) antibodies. ΔV_G_ of each step in Figure 7B was −2 mV, −3 mV, and −2 mV. The average values of ΔV_G_ between baseline/coupled enzyme PTS (M2-M1 in Figure 6A), anti-GST antibody/coupled enzyme PTS (M3-M2 in Figure 6A), and Anti-sulfo antibody/coupled enzyme PTS (M4-M2 in Figure 6A) are shown in Figure 6B. Figure 6A shows one of the four typical experiments, and the data of the other three experiments are shown in Appendix A. The data for the same experiments, except in the absence of TPST, are also shown (Figure 7A,B, and Appendix A, respectively).

We previously demonstrated that PTS can be produced using a traditional proteome chip on a silicon oxide substrate [29]. However, the detection and identification of PTS rely on fluorescence labeling and AFM for the proteome chip and silicon oxide substrate, respectively. In this report, we introduce a direct and label-free method for the detection of PTS, which may also be used for other PTMs. The pSNWFET method also enables that real-time detection of PTS and subsequent protein–protein interactions.

Figure 6A demonstrates the detection of PTS and its subsequent interaction with specific antibodies with I–V curves, a typical method used to characterize the electrical properties of a semiconductor chip. It is reasonable to expect that real-time observation of the biochemical events is possible if we fix the voltage and scan over time, as shown in Appendix A, in which AC conductance was measured to obtain electrical signals. The AC current signals were converted into AC voltage signals by a current preamplifier. With a frequency of 43 Hz, the differential conductance of the NWs at a fixed gate voltage was measured by an AC modulation signal. G_0_ is the conductance obtained from the anti-GST treatment as the baseline, and the changes in the conductance (G/G_0_ = G_anti-sulfotyrosine_/G_anti-GST_) were calculated at a fixed voltage.

In order to obtain reliable signals during biosensing, we performed a stability test of the device through pH profile, as shown in Appendix A. Before the biosensing procedure, the conductance would be continuously measured in the sensing buffer to confirm the stability of the signal. The real-time detection of PTS with an antibody demonstrated on three types of surface treatments and G/G_0_ are shown in Appendix A; the pSNWFET surface was unmodified (Appendix A), modified with nonsulfated PSGL-1 peptide (Appendix A), and modified with sulfated PSGL-1 (Appendix A). On the unmodified and nonsulfated PSGL-1 peptide-modified pSNWFET surface, negligible responses were obtained when the anti-GST antibody or the sulfotyrosine antibody were applied (Appendix A). On the sulfated PSGL-1 peptide-modified pSNWFET, only a specific antibody readily recognized the sulfated PSGL-1 peptide. This result confirmed that PST can be readily monitored, which is critical if a large array of semiconductor chips are assembled and a quick scanning of each sensing device is required. Appendix A illustrates the detection signal of protein–protein interactions on the pSNWFET surface. In this experiment, we used gold wire as pseudoreference electrode. Compared to a Ag/AgCl electrode, this type of electrode may be contaminated by the proteins in samples, resulting in different surface potential and drifting in continuous electrical measurement. In this research, the sensing and reaction were performed in a relatively clean system, including all the purified enzymes and proteins. Thus, we did not observe significant interference. However, we would switch to a Ag/AgCl electrode as reference electrode with a more complex system, such as the use of the biological sample.

### 3.6. Silicon NW Field-Effect Transistor as Next-Generation Biochip for Post-Translational Protein Modification

In the previous sections, we demonstrated that PTS and its subsequent interaction with a specific antibody could both be efficiently monitored by the pSNWFET. It also implies that the biological functions of PTS, generally induced after a protein–protein interaction, can also be monitored on a chip. Monitoring PTS has been difficult owing to the instability of the sulfated protein and the lack of a direct detection method in real time. We combined an in situ PTS system with a polysilicon NWFET-based sensor to directly monitor PTS formation. A similar method can be applied to other PTMs. Differently from other applications by the pSNWFET, this study aimed to develop the pSNWFET into a biochip rather than a biosensor. In a biosensor, the target is generally in the biological sample and is to be recognized by the immobilized probe on the device. However, in our study, the target is not in solution, but on the device, and is waiting to be identified. The idea is more like a biochip and thus we term that our development is for the next-generation biochip. In this study, we focus on the on-chip post-translational modification and detection for each step of the signal, and not on the real-time sensing of particular molecules in the biological sample. In this study, the sensitivity of the device was to be studied by the immobilized targets on the nanowire, but not the target in the sample solution as those of a biosensor. We thus term this a biochip rather than a biosensor. Appendix A is provided to demonstrate that the change of the status of the immobilized targets can be observed by electronic signals and the variation of the signal is dependent on the type of modification. In the poly-Si NWFET chip, this “sensitivity” is mainly dependent on the type of the modification rather than the concentration of the target. This approach may contribute significantly to the realization of PTMs and the investigation of related vital human diseases. Compared with the traditional biochip, the pSNWFET-based biochip may have the following advantages: Transistor devices compatible to biotargets such as biomolecules, virus, and bacteria can be prepared, which can provide a novel opportunity to efficiently monitor important biomarkers. The device can be efficiently prepared using state-of-the-art semiconductor technologies, which makes the platform very compact yet scalable and capable of maintaining high throughputs. The platform could be enabled by the on-chip integrated circuit and could continuously collect the kinetic responses between the target molecules and probes. The collected information could therefore, on-the-fly, be uploaded on to the Cloud for immediate data analysis. The pSNWFET-based diagnosis platform can be fully integrated and directly connected to the digital world, such as Internet of Things (IoT), Internet of Humans (IoH), AI, or cyber space. Semiconductor-based diagnostic technology can be utilized to monitor biochemical reactions and further monitor in-process biological reactions, which provide clear advantages for precision medicine and cloud-based medical care in the development of AI and big data applications.

Although pSNWFET has potential to be developed as a biosensing platform, there are still some challenges to be faced for real-world applications. First, in this study, the device we used was fabricated from National Nano Device Laboratories, which is an academic foundry. The reproducibility of the device is approximately up to 70%, however, recently, the processes have been transferred to commercial foundries (Hsinchu, Taiwan), and yield rate is up to 99% (yield is defined as the percentage of the semiconductor sensor that can be developed to a functional biosensor). Polycrystalline silicon has more structural defects than monocrystalline silicon, which may degrade the electrical properties of the device. To make sure that the device is functional during the entire period of biosensing procedures, every electrical measurement in the sensing process was repeated at least 5 times to ensure stability and no drifting of the electrical characteristics. Some fabrication steps, such as annealing in process was adopted to minimize the defects of the polysilicon. There are some factors that can cause an unexpected shift during the sensing process with real samples. For example, protein fouling is the gathering of protein aggregates on a surface, which may contaminate the sensors or the reference gate, resulting in a potential shift of electrical measurement. The washing step is critical to reduce contamination from protein adsorption. Surface modification with antifouling material is also a frequent remedy. Also, ionic strength of the real samples is typically high, which causes sensitivity loss due to background noise. To avoid unneeded noise, the proper concentration of the sensing buffer is required. A high ionic strength buffer can be used when the samples react with the probe on the sensor, but after the washing step, an appropriate ionic strength sensing buffer should be used to ensure the sensitivity. Many of the above variations are critical to biosensing methods for a reliable result. Past research indicates that potential solutions are available.

## 4. Conclusions

Through this study, we demonstrated a label-free, semiconductor-based pSNWFET for the electrical detection of PTM and, given the simple and efficient method for pSNWFET fabrication, the pSNWFET has great potential to be developed as a biochip/biosensor. Direct monitoring of protein modification is critical for many biomedical studies and disease treatments. The reported approach may complement the shortcomings of current methods, such as the lack of direct real-time detection and the requirement of a labeling step. In addition to PTS, NWFET-based PTM detection can be applied to other PTMs such as phosphorylation and glycosylation, which play important roles in several diseases and cellular communication. We expect this approach to contribute to the understanding of PTM-related human diseases and research in the future. With the fabrication of a CMOS-compatible pSNWFET in commercial foundries, integration with modern electronics can be achieved, and extensive applications in biomedical areas can be further explored. We expect that such a semiconductor-based biochip may soon become a powerful tool in biomedical research.

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
