# Peer review of "Silicon Nanowire Field-Effect Transistor as Biosensing Platforms for Post-Translational Modification"

_biosensors, 2020, doi:10.3390/bios10120213_

Round 1

Reviewer 1 Report

The experiments appear done with care and are overall adequately described. Nevertheless, some questions and comments remain that should be corrected prior publication:

  1. The authors appaer a too ambitious (in my opinion) towards the implementation of nanowire FETs in practical scenarious. To the best of my knowledge not a single NW device made it so far to the real market, which is due to variable reasons. Here I suggest that the authors try to keep a more realistic attitude in their article and at least re-think issues of fouling, cross-talk/-sensitivities, ref-electrode shift, and a sensitivity loss in high-ionic strength solutions, which is typically true for real samples.
  2. Many of the figures/labels and insets are too tiny (e.g. Fig. 2). And add labels to figures to show what a reader can see (e.g. Fig. 1). Please correct.
  3. Provide the contact material and the preparation method for the FETs
  4. Discuss the reproducibiliy of your devices.
  5. You used a gold-wire as liquid-gate electrode. Gold can easily get contaminated by proteins, which would result in a potential shift. Plesae discuss. 

Reviewer 2 Report

  1. Protein modification is quite critical to disease detection as well as its treatment. In this respect, this paper " Silicon nanowire field Effect Transistor as next Generation Biochip for Post Translational Modification" demonstrated polysilicon nanowire field effect transistor-based sensor as an effective sensor for electrical detection of protein-protein interactions. The authors have thoroughly explained the motivation of the subject, methods and measurements involved. The manuscript is well written and detailed. The authors have shown the comparison of protein-protein interaction results from traditional ELISA plate method and pSNWFET method. They have also used ELISA method as a first step to establish optimal biosensing condition. Electrical detection using pSNWFET showed high sensitivity and is a label free method. Large scale production of pSNWFET based biochip is possible as the process is CMOS compatible. However, I do have few questions and comments:

    1. Polysilicon nanowires have a lot of structural defects in them, which will degrade the electrical properties of the device. Does this influence electrical measurement results obtained by protein-protein interactions?
    2. Did you study the effect of NW diameter on the sensitivity of PTS? I think small diameter should show high sensitivity due to high surface by volume ratio.
    3. What is the reproducibility of the process?
    4. In Figure 4, 5A, 6A, 7A it is quite difficult to follow the symbols especially circles and rhombus looks very similar at this scale. Can you please add different colors to make it more visible and clearer?
    5. Line 292, Reference 34 should be in square brackets.

Reviewer 3 Report

In this manuscript, the authors present biosensors based on polysilicon nanowire field-effect transistors to directly monitor protein tyrosine sulfation formation which regulates physiological and pathological responses. The paper is well written in English and scientifically sound, however,  several minor and critical revisions are required before I recommend its publication on Biosensors.  

I wish to address that a preprint version of this paper is already available online without peer review doi:10.20944/preprints202011.0162.v1  

  1. Title modification: The present title “Silicon Nanowire Field-Effect Transistor as Next Generation Biochip for Post-Translational Modification” should be modified to remove improper nouns such as “Next Generation Biochip” since no complete prototype has been produced and the test was made on the SiFET platform. I would suggest to change it to: “Silicon Nanowire Field-Effect Transistor as Biosensing Platforms for Post-Translational Modification”
  1. Abstract: There are already several papers on this topic so I would suggest removing the phrase “To the best of our knowledge, this is the first study to describe in

situ PTS and its direct observation by using semiconductor devices.” on line 22-23 from the abstract at p.1.

  1. References: there are too many self-citations of the authors, I counted at least 32 %.The authors should limit their self-citation to only 5 papers maximum, without adding additional references.
  1. Images: the quality should be improved in resolution for all the presented schemes and figures and the graphs should be increased in size to make them more readable.
  2. The term “Nude” should be replaced with “bare” or “pristine” across with the whole paper and graph.
  3. The term “ultra-sensitive” should be removed from the text, title, paper, and abstract unless supported by experimental measurements and quantitative results.

Round 2

Reviewer 3 Report

In this manuscript, the authors present biosensors based on polysilicon nanowire field-effect transistors to directly monitor protein tyrosine sulfation formation which regulates physiological and pathological responses. The paper is well written in English and scientifically sound. The authors fulfilled the minor revisions required, hence I can now recommend its publication on Biosensors.